# Globally defining the effects of mutations in a picornavirus capsid

**Florian Mattenberger[1], Victor Latorre[1], Omer Tirosh[2], Adi Stern[2], Ron Geller[1]***

[1]Institute for Integrative Systems Biology, I2SysBio (Universitat de València-CSIC), Paterna, Spain; [2]The Shmunis School of Biomedicine and Cancer Research, Tel-Aviv University, Tel-Aviv, Israel

**Abstract** The capsids of non-enveloped viruses are highly multimeric and multifunctional protein assemblies that play key roles in viral biology and pathogenesis. Despite their importance, a comprehensive understanding of how mutations affect viral fitness across different structural and functional attributes of the capsid is lacking. To address this limitation, we globally define the effects of mutations across the capsid of a human picornavirus. Using this resource, we identify structural and sequence determinants that accurately predict mutational fitness effects, refine evolutionary analyses, and define the sequence specificity of key capsid-encoded motifs. Furthermore, capitalizing on the derived sequence requirements for capsid-encoded protease cleavage sites, we implement a bioinformatic approach for identifying novel host proteins targeted by viral proteases. Our findings represent the most comprehensive investigation of mutational fitness effects in a picornavirus capsid to date and illuminate important aspects of viral biology, evolution, and host interactions.

***For correspondence:**
ron.geller@uv.es

**Competing interests:** The authors declare that no competing interests exist.

## Introduction

The capsids of non-enveloped viruses are among the most complex of any viral protein. These highly multimeric structures must correctly assemble around the genome from numerous subunits, at times numbering in the hundreds, while avoiding aggregation (*Harrison, 2013*; *Hunter, 2013*; *Perlmutter and Hagan, 2015*). Moreover, the assembled structure must be both sufficiently stable to protect the viral genome during its transition between cells yet readily disassemble upon entry to initiate subsequent infections. For these functions to be achieved, viral capsids must encode the information for interacting with numerous cellular factors that are required to correctly fold and assemble around the genome (*Callaway et al., 2001*; *Fields et al., 2013*; *Geller et al., 2007*; *Jiang et al., 2014*; *Macejak and Sarnow, 1992*). Viral capsids also play key roles in pathogenesis, dictating host and cell tropism by encoding the determinants for binding cellular receptors (*Helenius, 2013*; *Rossmann et al., 2002*) and mediating escape from humoral immune responses (*Cifuente and Moratorio, 2019*; *Heise and Virgin, 2013*). As a result, viral capsids show the highest evolutionary rates among viral proteins.

The picornaviruses constitute a large group of single-stranded, positive-sense RNA viruses and include several pathogens of significant medical and economic impact (*Racaniello, 2013*). Their relative simplicity and ease of culture have made picornaviruses important models for understanding virus biology. Among the many breakthroughs achieved with these viruses was the determination of the first high-resolution structure of the capsid of an animal virus, making the picornavirus capsid the prototypical non-enveloped, icosahedral viral capsid (*Racaniello, 2013*). Picornavirus capsid genesis initiates with the co-translational release of the P1 capsid precursor protein from the viral polyprotein via the proteolytic activity of the viral encoded 2A protease (*Jiang et al., 2014*; *Racaniello, 2013*). Subsequently, the viral encoded 3CD protease (3CD^pro) cleaves the P1 capsid precursor to liberate three capsid proteins (VP0, VP3, and VP1), generating the capsid protomer.

**eLife digest** A virus is made up of genetic material that is encased with a protective protein coat called the capsid. The capsid also helps the virus to infect host cells by binding to the host receptor proteins and releasing its genetic material. Inside the cell, the virus hitchhikes the infected cell's machinery to grow or replicate its own genetic material.

Viral capsids are the main target of the host's defence system, and therefore, continuously change in an attempt to escape the immune system by introducing alterations (known as mutations) into the genes encoding viral capsid proteins. Mutations occur randomly, and so while some changes to the viral capsid might confer an advantage, others may have no effect at all, or even weaken the virus.

To better understand the effect of capsid mutations on the virus' ability to infect host cells, Mattenberger et al. studied the Coxsackievirus B3, which is linked to heart problems and acute heart failure in humans. The researchers analysed around 90% of possible amino acid mutations (over 14,800 mutations) and correlated each mutation to how it influenced the virus' ability to replicate in human cells grown in the laboratory.

Based on these results, Mattenberger et al. developed a computer model to predict how a particular mutation might affect the virus. The analysis also identified specific amino acid sequences of capsid proteins that are essential for certain tasks, such as building the capsid. It also included an analysis of sequences in the capsid that allow it to be recognized by another viral protein, which cuts the capsid proteins into the right size from a larger precursor. By looking for similar sequences in human genes, the researchers identified several ones that the virus may attack and inactivate to support its own replication.

These findings may help identify potential drug targets to develop new antiviral therapies. For example, proteins of the capsid that are less likely to mutate will provide a better target as they lower the possibility of the virus to become resistant to the treatment. They also highlight new proteins in human cells that could potentially block the virus in cells.

Five protomers then assemble to form the pentamer, twelve of which assemble around the viral genome to yield the virion. Finally, in some picornaviruses, VP0 is further cleaved into two subunits, VP4 and VP2, following genomic encapsidation to generate the infectious, 240 subunit particles (*Jiang et al., 2014*; *Racaniello, 2013*). Work over the years has identified numerous host factors that help support capsid formation (*Corbic Ramljak et al., 2018*; *Geller et al., 2007*; *Macejak and Sarnow, 1992*; *Qing et al., 2014*; *Thibaut et al., 2014*), defined antibody neutralization sites (*Cifuente and Moratorio, 2019*), and identified numerous host receptors for many members of this viral family (*Rossmann et al., 2002*).

Despite significant progress in understanding the structure and function of picornavirus capsids, a comprehensive understanding of how mutations affect viral fitness across different structural and functional attributes is lacking. To address this, we perform a comprehensive analysis of mutational fitness effects (MFE) across the complete capsid region of the human picornavirus coxsackievirus B3 (CVB3), analyzing >90% of all possible single amino acid mutations. Furthermore, using these data, we develop models to predict the effect of mutations with high accuracy from available sequence and structural information, improve evolutionary analyses of CVB3, and define the sequence preferences of several viral encoded motifs. Finally, we use the information obtained in our dataset for the sequence requirements of capsid-encoded 3CD protease cleavage sites to identify host targets of this viral protease. Overall, our data comprise the most comprehensive survey of MFE effects in a picornavirus capsid to date and provide important insights into virus biology, evolution, and interaction with the host.

## Results

### Deep mutational scanning of a CVB3 capsid

To generate CVB3 libraries encoding a large amount of diversity in the capsid region, we used a codon-level PCR mutagenesis method (*Bloom, 2014*). The mutagenesis protocol was performed on

the capsid precursor region P1 in triplicate to generate three independent mutagenized libraries (Mut Library 1–3; *Figure 1A*). From these, three independent viral populations (Mut Virus 1–3) were derived by electroporation of in vitro transcribed viral RNA into HeLa-H1 cells (*Figure 1A*). High-fidelity next-generation sequencing (*Schmitt et al., 2012*) was then used to analyze the mutagenized libraries and resulting viruses, unmutagenized virus populations (WT virus 1–2), as well as controls for errors occurring during PCR (PCR) and reverse transcription (RT-PCR). High coverage was obtained for all samples (>$10^6$ per codon across all experimental conditions and >$6.5 \times 10^5$ for the controls; *Supplementary file 2*). Due to the high rate of single mutations within codons observed in the RT-PCR control compared to the mutagenized virus populations (*Supplementary file 2*), all single mutants were omitted from our analysis to increase the signal-to-noise ratio. While this resulted in an inability to analyze 83.4% of synonymous codons in the capsid region (1746/2094), only 2.8% of non-synonymous mutations were lost to analysis (458/16,169). Upon removing single mutations within codons, we obtained a large signal-to-noise ratio in the average mutation rate of 510× (range 449–572) and 245× (range 174–285) for the mutagenized libraries and viruses, respectively, compared to their error controls (*Figure 1B* and *Supplementary file 2*). On average, 0.9 (range 0.8–1.02) codon mutations were observed per genome, which was in agreement with Sanger sequencing of 59 clones (range 18–23 per library; *Figure 1—figure supplement 1* and *Supplementary file 3*). As expected, the rate of stop codons, which should be invariably lethal in the CVB3 capsid, decreased significantly following growth in cells to <0.5% of that observed in the corresponding mutagenized libraries (p<0.005 by paired t-test on log-transformed data; *Supplementary file 2*). No major bias was observed in the position within a codon where mutations were observed (*Figure 1—figure supplement 2*) or in the type of mutation (*Figure 1—figure supplement 2*), except for the WT virus, which had a high rate of A to G transitions in the two independent replicates analyzed. Of all 16,169 possible amino acid mutations in the capsid region (851 AA × 19 AA mutation = 16,169), a total of 14,839 amino acid mutations were commonly observed in all three mutagenized libraries, representing a 91.8% of all possible amino acid mutations in the capsid region, allowing us to globally assess the effects of the vast majority of amino acid mutations on the capsid (*Figure 1C*).

## MFE across the CVB3 capsid

We next derived the MFE of each observed mutation by examining how its frequency changed relative to that of the WT sequence following growth in cells. The preferences for the different amino acids at each position (amino acid preferences [*Bloom, 2015*]) showed a high correlation between biological replicates (Spearman's ρ > 0.83; *Figure 2—figure supplement 1* and *Supplementary file 4* MFE). Overall, most mutations in the capsid were deleterious to growth in cell culture, with only 1.2% of mutations increasing fitness relative to the WT amino acid (*Figure 2A* and *Supplementary file 4*; Interactive heatmap available at https://rgellerlab.github.io/CVB3_capsid_DMS_Interactive_Heatmap/). Hotspots where mutations were tolerated were observed at several regions across the capsid (*Figure 2A*). These hotspots largely overlapped with highly variable regions in natural sequences, as measured by Shannon entropy in the enterovirus B family, indicating that lab measured MFE reflect natural evolutionary processes (*Figure 2A*, top). Indeed, a strong correlation was observed between the average MFE observed at each site and sequence variability for the enterovirus B genus (Spearman's ρ = 0.59, p<$10^{-16}$; *Figure 2B*). Similarly, antibody neutralization sites overlapped with hotspots for mutations (*Figure 2A*, top), with individual mutations in antibody neutralization sites showing lower MFE (p<$10^{-16}$ by Mann–Whitney test; *Figure 2C*). As expected, mutations were also less deleterious in loops compared to β-strands (p<$10^{-16}$ by Mann–Whitney test; *Figure 2D*), at surface residues compared to core residues (p<$10^{-16}$ by Mann–Whitney test; *Figure 2E*), and for mutations predicted to be destabilizing or aggregation-prone (p<$10^{-16}$ by Mann–Whitney test for both; *Figure 2F*). Importantly, independent validation of the MFE of 10 different mutants using a sensitive qPCR-based competition assay (*Moratorio et al., 2017*) showed a strong correlation with the deep mutational scanning (DMS) results (Spearman's ρ = 0.9, p<0.001; *Figure 2G* and *Supplementary file 5*). It is important to note that laboratory-measured MFE may not always reflect those in nature due to differences in the environments.

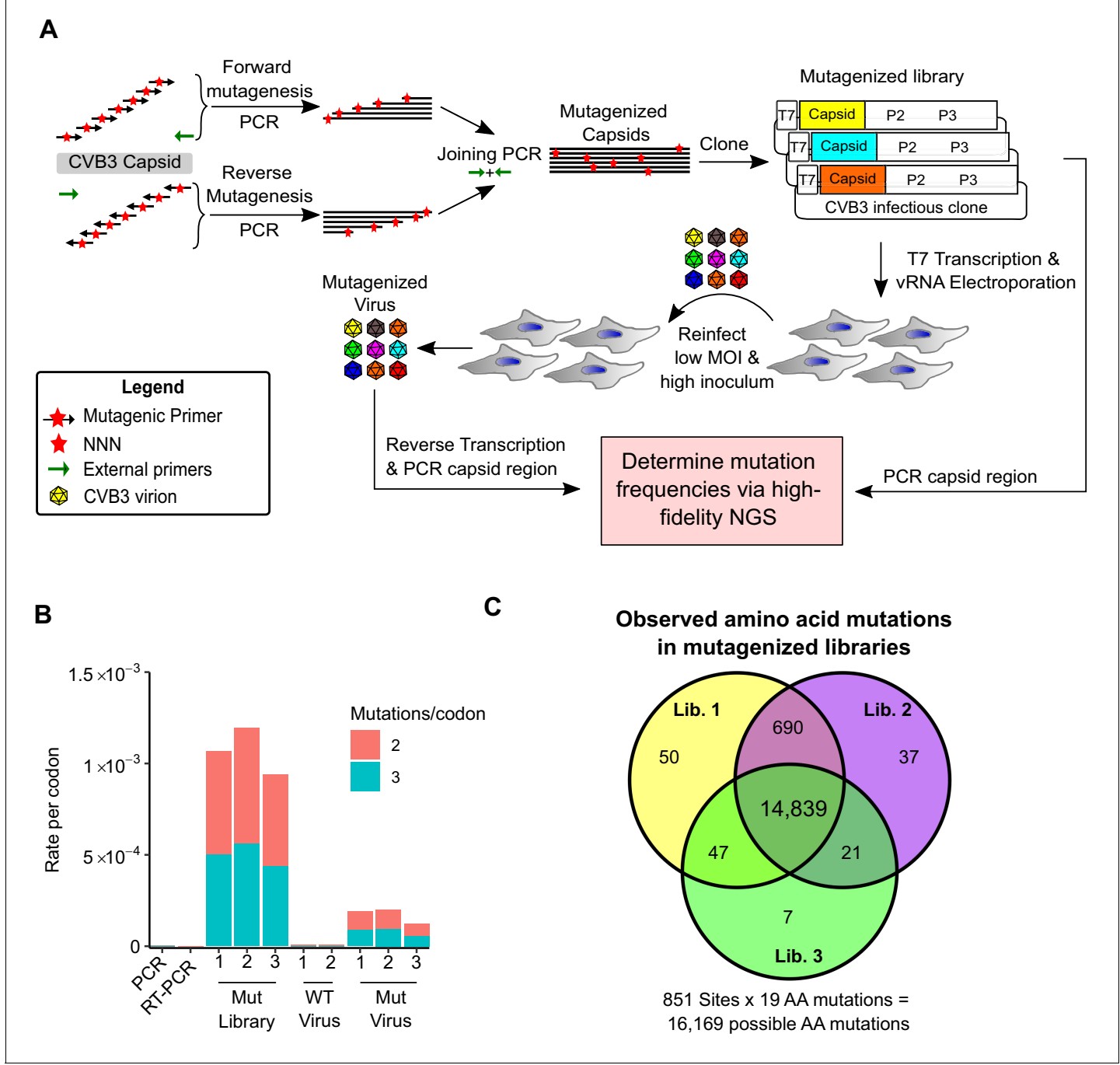

**Figure 1.** Deep mutational scanning (DMS) of the CVB3 capsid. (A) Overview of the deep mutational scanning experimental approach. A mutagenesis PCR was used to introduce all possible single amino acid mutations across the CVB3 capsid region (Mut Library 1–3). Viral genomic RNA (vRNA) produced from the mutant libraries was then electroporated into cells to generate high diversity CVB3 populations (Mut Virus 1–3). The frequency of each mutation relative to the WT amino acid was then determined in both the mutagenized libraries and the resulting virus populations via high-fidelity duplex sequencing. (B) The average rate of double or triple mutations per codon observed in the mutagenized libraries (Mut Library 1–3), the resulting mutagenized virus (Mut Virus 1–3), as well as controls for the error rate of the amplification and sequencing process (PCR and RT-PCR) or the WT unmutagenized virus (WT Virus 1–2). Single mutations per codon were omitted from the analysis to increase the signal-to-noise ratio. (C) Venn diagram showing the number of amino acid mutations observed in the mutagenized libraries. MOI: multiplicity of infection. NGS: next-generation sequencing. The online version of this article includes the following figure supplement(s) for figure 1:

**Figure supplement 1.** Sanger analysis of DMS libraries.
**Figure supplement 2.** Results of high-fidelity duplex sequencing.

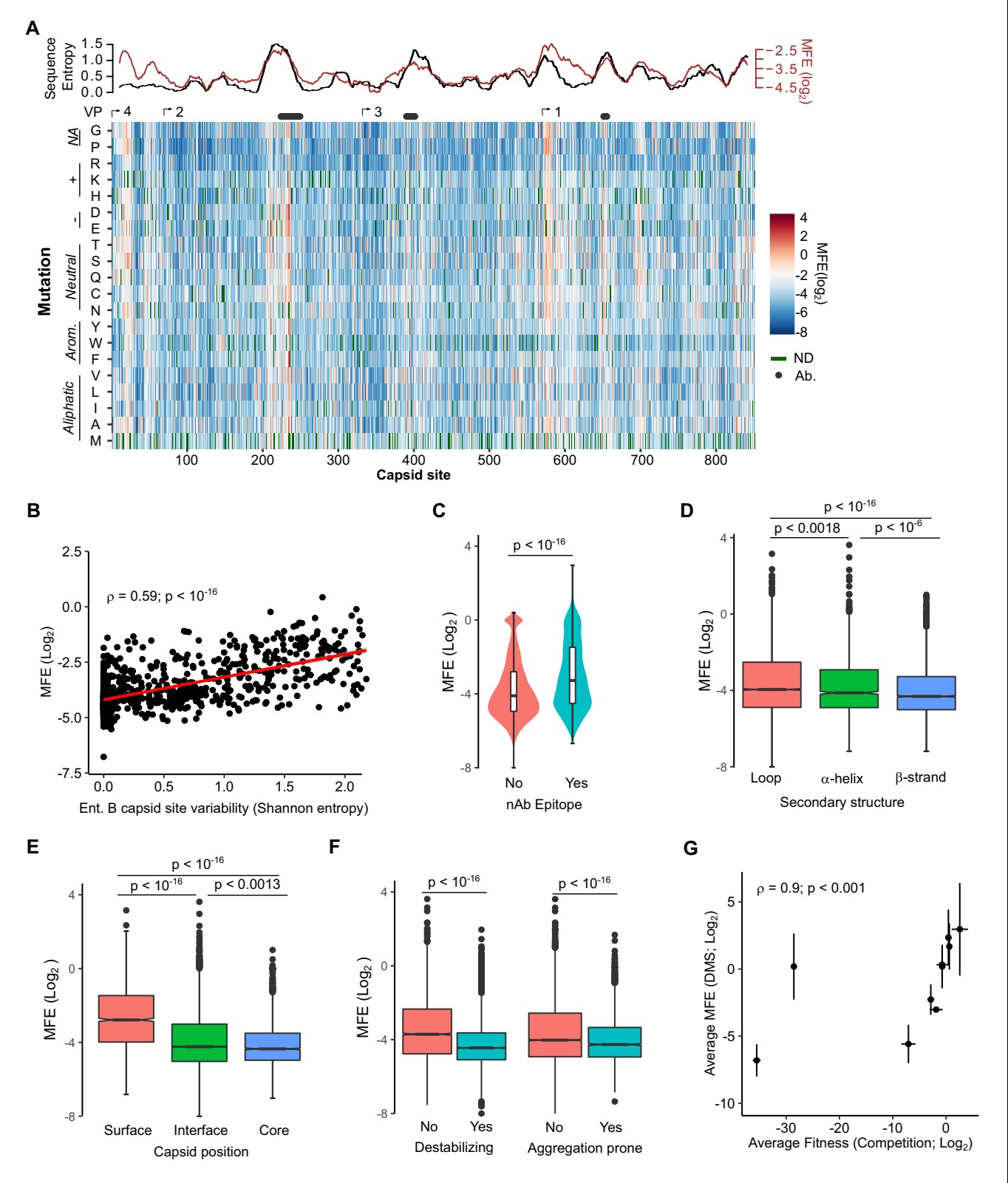

**Figure 2.** Mutational fitness effects (MFE) across the CVB3 capsid and their correlation with structural, evolutionary, and immunological attributes. (**A**) Overview of the MFE observed across the CVB3 capsid. Bottom: A heatmap representing the MFE of all mutations observed at each capsid site. Green indicates no data available (ND), and the positions of the mature viral proteins (VP1–4) or antibody neutralization sites (nAb) are indicated above. Top: A 21 amino acid sliding window analysis of the average sequence variation in enterovirus B genomes (Shannon entropy; black line) or a 21 amino acid

*Figure 2 continued on next page*

*Figure 2 continued*

sliding window of the average MFE observed at each capsid site (red line). (**B**) Correlation between the average MFE observed at each capsid site and variation in enterovirus B sequence alignments (Shannon entropy). (**C**) Violin plot of MFE in antibody neutralization sites versus other capsid sites. (**D–F**) Boxplots of MFE as a function of secondary structure (**D**), position in the capsid (**E**), or the predicted effect of mutations on stability or aggregation propensity (**F**). (**G**) Validation of the MFE obtained by DMS using a competition assay. For each mutant, the average and standard deviation of the MFE obtained by DMS (n = 3) is plotted against the average and standard deviation of the fitness derived using the competition assay (n = 4). A two-sided Mann–Whitney test was used for two- category comparisons.

The online version of this article includes the following figure supplement(s) for figure 2:

**Figure supplement 1.** Correlation of amino acid preferences observed in experimental replicates.

## Prediction of MFE from available structural and sequence information

As MFE correlated with natural sequence variation and different structural features of the capsid (*Figure 2*), we next investigated if MFE could be predicted from available structural and sequence information. For this, we obtained a dataset of 52 parameters, including structural information derived from the crystal structure of the CVB3 capsid (PDB:4GB3), amino acid properties, and natural variation in available enterovirus sequences (Shannon entropy), and predicted the effects of mutation on stability and aggregation propensity using FoldX (*Schymkowitz et al., 2005*) and TANGO (*Fernandez-Escamilla et al., 2004*), respectively (*Supplementary file 6*). We then employed a random forest algorithm to identify the parameters that can best predict MFE, limiting our analysis to sites that present in the crystal structure and where mutations were observed in at least two replicates to improve accuracy (total of 9685 mutations). Overall, a model trained on 70% of the dataset was able to predict the remaining 30% of the data (2905 mutations) with high accuracy (Spearman's $\rho > 0.75$, Pearson's r = 0.76; $p<10^{-16}$; *Figure 3—figure supplement 1*). Surprisingly, a random forest model trained on the top five predictors alone showed similar accuracy (Spearman's $\rho = 0.73$, Pearson's r = 0.73; $p<10^{-16}$; *Figure 3B*). Excluding natural sequence variation, amino acid identity, or structural attributes reduced model predictability significantly (>20%; data not shown), suggesting

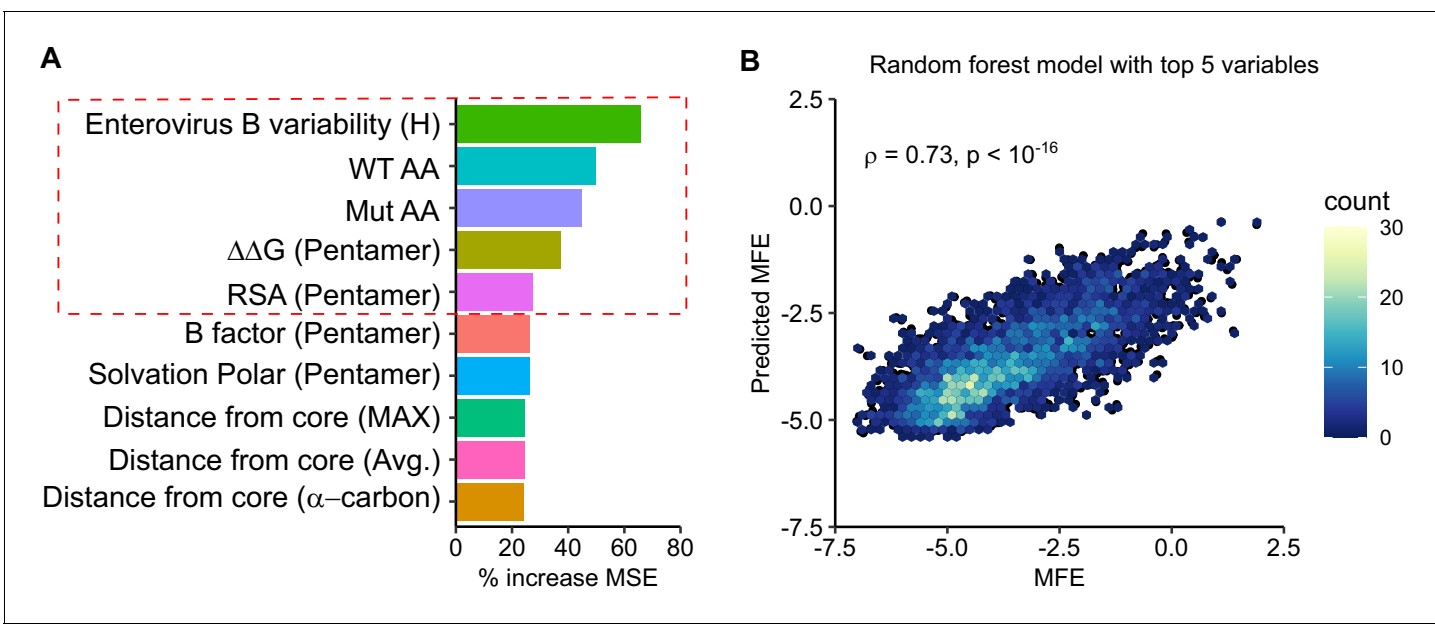

**Figure 3.** Prediction of MFE based on structural and sequence information. (**A**) The top 10 predictors identified in a random forest model for explaining MFE in the CVB3 capsid based on the percent of mean squared error (MSE) increase. (**B**) Hexagonal plot showing the correlation between MFE predicted using a random forest algorithm trained on the top five variables versus observed MFE. The random forest model was trained on 70% of the data and then tested on the remaining 30% (shown). RSA, relative surface area.

The online version of this article includes the following figure supplement(s) for figure 3:

**Figure supplement 1.** Prediction of mutational fitness effects using random forest or linear models.

a combination of evolutionary, sequence, and structural information best explains MFE. Using an alternative approach, we were able to predict the data with slightly lower accuracy using a linear model with the same five predictors (p<10$^{-16}$, Spearman's ρ = 0.67, Pearson's r = 0.67; *Figure 3—figure supplement 1*). Together, these results suggest that the prediction of MFE in the CVB3 capsid can be achieved at relatively high accuracy based on available structural and sequence information. Due to the high conservation of capsid structure in picornaviruses, as well as the availability of numerous capsid sequences and structures, these findings are likely generalizable to related picornaviruses.

## Experimentally measured MFE inform of natural evolutionary processes

We next examined if our experimentally measured MFE could improve phylogenetic models of CVB3 evolution by incorporating site-specific amino acid preferences using PhyDMS (*Hilton et al., 2017*). Indeed, significant improvement in model fit was observed (*Table 1* PHY; p<10$^{-16}$ using a log-likelihood test compared to non-site-specific codon models), supporting the relevance of our results to understanding evolutionary processes in nature. Nevertheless, selection in nature was significantly more stringent than in the lab (β = 2.18), indicating the presence of additional selection pressures. As laboratory conditions lack selection from antibodies, we used the sum of the absolute differential selection observed at each site (*Bloom, 2017*) to examine whether known antibody neutralization sites show differential selection between the two environments (*Supplementary file 7*). Indeed, antibody neutralization sites showed significantly higher differential selection values compared to other residues (p<10$^{-6}$ by Mann–Whitney test; *Figure 4A*). Moreover, the three sites showing the strongest overall differential selection were found in known antibody neutralization sites: positions 226 and 242 in the EF loop (residues 157 and 173 of VP2) and position 650 in the BC loop (residue 80 of VP1; *Figure 4B–D* and *Supplementary file 7*). In summary, incorporation of experimentally derived amino acid preferences into phylogenetic analyses significantly improved model fit and identified residues in antibody neutralization sites that show differential selection, suggesting these may play important roles in immune evasion in vivo.

## Insights into capsid-encoded motifs: myristoylation and protease cleavage

Picornavirus capsids undergo a complex assembly path to generate the infectious particle. These include myristoylation, cleavage by the viral proteases 2A and 3CD$^{pro}$, as well as interaction with cellular chaperones and glutathione (*Corbic Ramljak et al., 2018*; *Geller et al., 2007*; *Jiang et al., 2014*; *Qing et al., 2014*; *Thibaut et al., 2014*; *Figure 5A*). Having obtained a comprehensive dataset for MFE across the capsid, we next examined the sequence requirements for several of these capsid-encoded motifs. Specifically, myristoylation of the N-terminal glycine is essential for virion assembly (*Corbic Ramljak et al., 2018*). In agreement with this, the N-terminal glycine in the CVB3 capsid showed the strongest average fitness cost upon mutation in the capsid (*Figure 4—figure supplement 1* and *Supplementary file 4*). The remaining sites in the myristoylation motif agreed with the canonical myristoylation motif in cellular proteins (Prosite pattern PDOC00008) (*Bologna et al., 2004*), albeit with increased selectivity at three of the six positions (*Figure 4—figure supplement 1*). On the other hand, a conserved WCPRP motif in the C-terminal region of VP1 that was shown to be important for 3CD$^{pro}$ cleavage of the related foot and mouth disease virus capsid (FDMV; YCPRP motif) (*Kristensen and Belsham, 2019*) was found to be intolerant to mutations

**Table 1.** Incorporation of DMS results in evolutionary models better describes natural CVB3 evolution compared to standard codon models.

| Model | ΔAIC | Log-likelihood | Parameters | Parameter values |
|---|---|---|---|---|
| ExpCM | 0.00 | −14,580.51 | 6 | Beta = 2.18, kappa = 7.47, omega = 0.16 |
| Goldman-Yang M5 | 4187.56 | −16,668.29 | 12 | Alpha_omega = 0.30, beta_omega = 10.00, kappa = 7.15 |
| Averaged ExpCM | 4303.74 | −16,732.38 | 6 | Beta = 0.61, kappa = 7.55, omega = 0.02 |
| Goldman-Yang M0 | 4371.26 | −16,761.14 | 11 | Kappa = 7.14, omega = 0.02 |

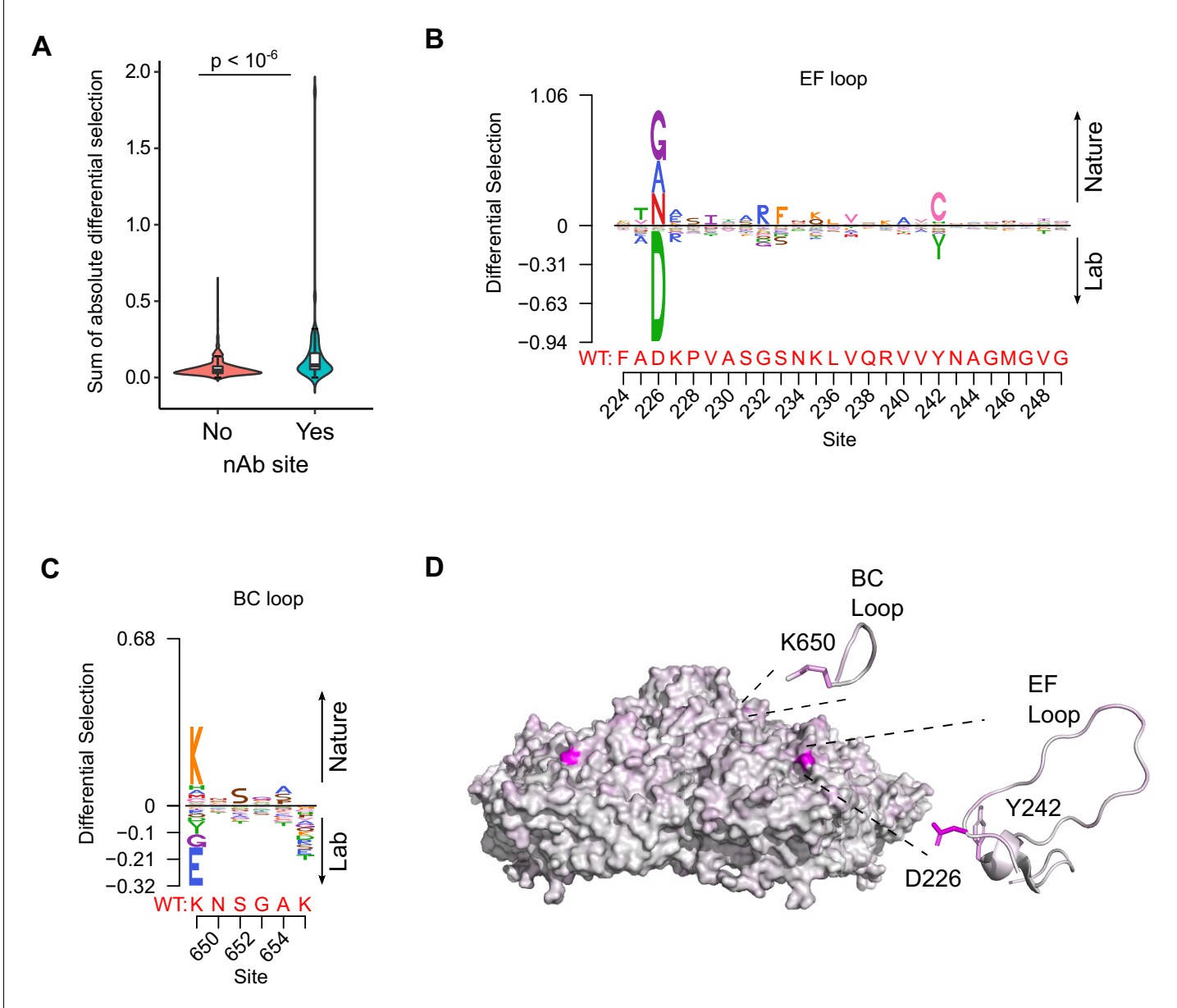

**Figure 4.** Antibody neutralization sites show differential selection between laboratory conditions and nature. (**A**) Violin plot showing the sum of absolute differential selection observed at capsid sites comprising antibody neutralization epitopes (nAb) versus all other capsid sites. (**B–C**) Logoplots showing the observed differential selection of sites in the EF loop or BC loop. The WT sequence is indicated in red. (**D**) The CVB3 capsid pentamer (PDB:4GB3), colored according to the amount of differential selection. The BC and EF loops are shown next to the structure together with the side chains for sites showing the highest differential selection.

The online version of this article includes the following figure supplement(s) for figure 4:

**Figure supplement 1.** Sequence preferences of capsid-encoded motifs.

compared to other capsid residues (p<0.05 versus all other positions by Mann–Whitney test; sites 815–819 in CVB3). Moreover, within this motif, the sites showing the highest average fitness cost in our DMS dataset were identical to analogous positions in FMDV that resulted in a loss of viability upon mutation to alanine (*Figure 4—figure supplement 1*; *Kristensen and Belsham, 2019*), highlighting the conservation of this motif across different picornaviruses.

The viral 3C protease (3C$^{pro}$) cleaves the picornavirus capsid at two conserved glutamine–glycine (QG) pairs to liberate the viral capsid proteins VP0, VP3, and VP1 (*Figure 5A*). Previous work has

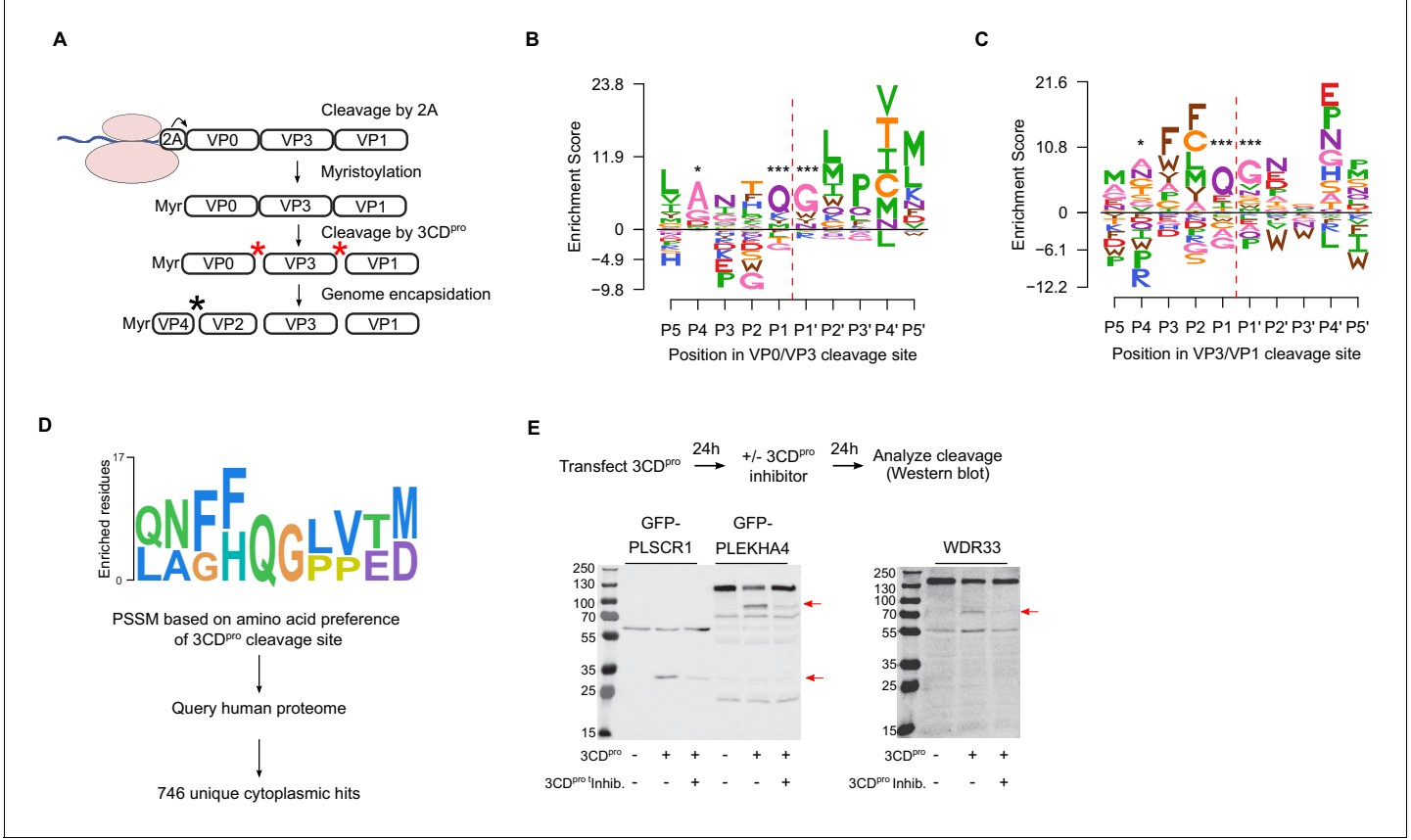

**Figure 5.** Sequence preference of capsid 3CD^pro cleavage sites and their use for the identification of novel cellular targets of the viral protease. (**A**) Overview of the CVB3 capsid maturation pathway. The CVB3 capsid precursor P1 is co-translationally cleaved by the viral 2A protease. P1 is then myristoylated and cleaved by the viral 3CD^pro to generate the capsid proteins VP0, VP3, and VP1. Finally, upon assembly and genome encapsidation, VP0 is further cleaved into VP4 and VP2 in a protease-independent manner to generate the mature capsid. Red and black asterisks indicated 3CD^pro or protease-independent cleavage events, respectively. (**B,C**) Logoplots showing amino acid preferences for the 10 amino acid regions spanning the 3CD^pro cleavage sites (P1–P'1) of both VP0/VP3 and VP3/VP1 in the DMS dataset. (**D**) Overview of the bioinformatic pipeline for identification of novel 3CD^pro cellular targets using the amino acid preferences for the capsid cleavage sites from our DMS study. A position-specific scoring matrix (PSSM) was generated based on the amino acid preferences for the 10 amino acid regions spanning the two 3CD^pro cleavages sites. This PSSM was then used to query the human genome for potential cellular targets, and non-cytoplasmic proteins were filtered out, yielding 746 proteins. (**E**) The cellular proteins PLSCR1, PLEKHA4, and WDR33 are cleaved by 3CD^pro. Western blot analysis of cells cotransfected with 3CD^pro and GFP-PLSCR1 or GFP-PLEKHA4 and probed with a GFP antibody or transfected with 3CD^pro and probed using a WDR33 antibody. When indicated, the 3CD^pro inhibitor rupintrivir was included to ensure cleavage was mediated by the viral protease. Red arrows indicate cleavage products of the expected size (GFP-PLSCR1 full length = 64 kDa, cleaved N-terminus = 36 kDa; GFP-PLEKHA4 full length = 118 kDa, cleaved N-terminus = 72 kDa; WDR33 full length = 146 kDa, cleaved N-terminus = 72 kDa). *p<0.05, ***p<0.001.

The online version of this article includes the following figure supplement(s) for figure 5:

**Figure supplement 1.** Evaluation of select hits identified as potential 3CD^pro target proteins.

---

defined the sequence specificity of several picornavirus 3C^pro enzymes by examining both natural sequence variation and in vitro cleavage assays using synthetic peptides (*Laitinen et al., 2016*). However, unlike other 3C^pro-mediated cleavage events in the viral polyprotein, the capsid is only efficiently cleaved by the precursor protein 3CD^pro (*Ypma-Wong et al., 1988*). To gain insights into the sequence specificity of 3CD^pro, we examined the amino acid preferences for a 10 amino acid region surrounding the protease cleavage site (P5–P5'). As expected based on the known specificity of the 3C protease (*Laitinen et al., 2016*), a strong preference for the presence of QG was observed at both 3CD^pro cleavage sites in our dataset (positions P1 and P1' in the cleavage site; *Figure 5B,C*). Interestingly, significant correlation in amino acid preferences between the two cleavage sites was observed only at P1–P1' (Pearson's ρ > 0.99, p<10^−16) and P4 (Pearson's ρ > 0.49, p<0.05), as was the case in the enterovirus B alignments (Pearson's ρ > 0.84 and p<10^−6 for positions P4, P1, and

P'1; data not shown). Hence, the low agreement in amino acid preferences observed for most positions across the two 3CD^pro cleavage sites suggests cleavage is strongly dictated by positions P4, P1, and P1'.

## Identification of 3CD^pro cellular targets based on the sequence preferences of capsid-encoded protease cleavage sites

In addition to cleaving the viral polyprotein, the picornavirus proteases cleave cellular factors to facilitate viral replication, including both antiviral factors and cellular factors that favor viral IRES-driven translation mechanism over cellular cap-dependent translation (e.g. DDX58, eIF4G, and PABP) (*Laitinen et al., 2016*; *Sun et al., 2016*). As the canonical 3C/3CD^pro QG cleavage site occurs on average 1.6 times per protein in the human proteome (~33,000,000 times), we sought to examine whether the rich dataset we obtained for the amino acid preferences of the capsid 3CD^pro cleavage sites can be used to identify novel cellular factors that are targeted by the viral protease. Specifically, a position-specific score matrix (PSSM) was generated for the 10 amino acid regions spanning the two protease cleavage sites in the CVB3 capsid (P5–P5') based on the amino acid preferences identified in our study (*Figure 5D*). This PSSM was then used to query the human proteome for potential cleavage sites, yielding a total of 746 cytoplasmic proteins (*Figure 5D*; *Supplementary file 8*). Eleven cellular factors that are known to be cleaved during enterovirus infection were identified using this approach, including the viral sensor Probable ATP-dependent RNA helicase DDX58 (RIG1), the immune transcription factors p65 (RELA) and interferon regulatory factor 7 (IRF7), and polyadenylate-binding protein 1 (PABPC1), an important factor in translation initiation and mRNA stability (*Supplementary file 8*; *Jagdeo et al., 2018*; *Laitinen et al., 2016*).

To evaluate whether our approach can identify novel cellular targets for the viral protease, we examined the ability of 3CD^pro to cleave eight different proteins found in the data set, focusing on those with cellular functions of potential relevance to CVB3 biology and which could be readily detected in our cell culture assay (e.g. availability of antibodies or tagged-variants, cleavage fragments of observable size, and high expression level). These included four interferon-inducible proteins (Pleckstrin homology domain containing A4, PLEKHA4; phospholipid scramblase 1, PLSCR1; NOD-like receptor family CARD domain containing 5, NLRC5; zinc finger CCCH-type containing, antiviral 1, ZC3HAV1) and four proteins involved in various cellular functions, namely apoptosis (MAGE family member D1, MAGED1), RNA processing (WD repeat domain 33, WDR33), and vesicle transport (cyclin G-associated kinase, GAK; tumor susceptibility 101, TSG101). Of these, three proteins were cleaved upon expression of the viral protease to generate fragments of the expected size (PLSCR1, PLEKHA4, and WDR33; *Figure 5E* and *Supplementary file 8*). Of note, while WDR33 was predicted to harbor two potential cleavage sites, only a single cleavage event was observed. Treatment with a specific 3CD^pro inhibitor, rupintrivir (*Dragovich et al., 1999*), blocked the cleavage of these proteins, indicating the effect was due to the viral protease (*Figure 5D*). In contrast, five of the proteins were found to not be cleaved upon 3CD^pro expression, suggesting additional determinants are involved in the cleavage of host factors (*Figure 5—figure supplement 1*). Hence, our approach correctly identified 30% of the predicted cleavage sites (three of the nine different cleavage sites), indicating a strong enrichment of cellular targets of the 3CD^pro in the dataset.

## Discussion

The picornavirus capsid is a highly complex structure that plays key roles in viral biology and pathogenesis. In the current study, we employ a comprehensive approach to define the effects of single amino acid mutations in the CVB3 capsid, measuring the effects of >90% of all possible mutations. We find that most mutations in the capsid are deleterious to growth in cell culture, with very few mutations showing higher fitness than the WT sequence (1.2% of all mutations). Similar results have been reported in other non-enveloped capsid proteins (*Acevedo et al., 2014*; *Hartman et al., 2018*; *Ogden et al., 2019*) as well as non-capsid viral proteins (*Ashenberg et al., 2017*; *Bloom, 2014*; *Doud and Bloom, 2016*; *Du et al., 2016*; *Haddox et al., 2016*; *Hom et al., 2019*; *Thyagarajan and Bloom, 2014*; *Wu et al., 2015*). In light of these results, it is likely that the large population sizes of RNA viruses help maintain viral fitness in the face of high mutation rates and strong mutational fitness costs. It is important to note that the effect of a particular mutation on

fitness observed under laboratory conditions may not always reflect its effect in nature due to inherent differences between these two environments.

Investigation of the factors that influence MFE in the capsid revealed a strong correlation with various structural and functional attributes. These included computationally predicted effects on stability and aggregation propensity, secondary structure, and surface exposure (*Figure 2*). Surprisingly, we find that MFE can be predicted with relatively high accuracy using only five parameters: natural sequence variation, the identity of the original and mutant amino acids, the predicted effect on protein stability, and relative solvent accessibility (*Figure 3*). A recent study examined the ability of 46 different variant effect prediction tools to predict MFE from 31 different DMS datasets of both viral and non-viral proteins (*Livesey and Marsh, 2020*). Overall, viral proteins showed the lowest predictability (Spearman's correlation of <0.5). In contrast, we were able to predict MFE using a random forest model using these above-mentioned five parameters with an accuracy similar to the best prediction obtained in this analysis for any viral or non-viral protein (Pearson's r = 0.73; Spearman's ρ = 0.73; *Figure 3B*). Interestingly, SNAP2 (*Hecht et al., 2015*), a neural network-based classifier of mutational effects that was shown to correlate well with MFE in other studies (*Gray et al., 2018*; *Livesey and Marsh, 2020*; *Reeb et al., 2020*), correlated poorly with our data ($R^2$ = −0.26). Overall, considering the relative conservation of capsid structure in picornaviruses as well as the availability of both capsid sequences and high-resolution structures for numerous members of this family, it is likely that these findings can be extrapolated to additional picornaviruses.

Incorporating site-specific amino acid preferences obtained from our DMS results into phylogenetic models was found to significantly improve model accuracy. This has been observed in DMS studies with other RNA viruses (*Bloom, 2017*; *Doud and Bloom, 2016*; *Haddox et al., 2018*) and indicated that our laboratory-measured MFE capture additional information that cannot be obtained from sequence analysis alone. In addition, this approach allowed us to assess which sites show differential selection patterns as a result of the distinct environments encountered in nature and the laboratory. As expected, pressure from the adaptive immune system was found to be the major difference between these environments, with residues in antibody neutralization sites showing higher differential selection compared to other sites in the capsid (*Figure 4A*). Moreover, the sites showing the highest degree of differential selection were found in known antibody neutralization sites (*Figure 4B–D*). However, why these particular residues within antibody neutralization sites show differential selection, while others do not, remain to be elucidated. It has been shown that one, or a few, sites within antibody binding regions can have strong effects on escape from antibody neutralization (*Lee et al., 2019*), potentially explaining these findings. Interestingly, while the top three sites showing differential selection were in antibody neutralization sites, the mutation showing the fourth-highest differential selection was found in the HI loop of VP1. While not classically considered an antibody epitope, this loop has been shown to interact with an antibody fragment in the picornavirus coxsackievirus A6 (*Xu et al., 2017*), is known to mediate receptor binding in different picornaviruses (*Belnap et al., 2000*; *Xing et al., 2000*), and to interact with host cyclophilin A to facilitate uncoating (*Qing et al., 2014*). Whether these factors or others are responsible for the observed differential selection remains to be elucidated.

The CVB3 capsid encodes the information for directing myristoylation, protease cleavage, and interaction with host factors. We took advantage of our data to examine the sequence specificity and mutational tolerance of several known capsid-encoded motifs. First, we examined the amino acid preferences of the CVB3 capsid myristoylation motif. We observe a strong correlation with the canonical myristoylation pattern (Prosite pattern PDOC00008), although with greater intolerance to mutations in three of the six residues in the capsid (*Figure 4—figure supplement 1*). This is likely to stem from additional constraints imposed by capsid structure. On the other hand, we examined the amino acid preference of a conserved motif in VP1 that is required for 3CD^pro-mediated cleavage of picornavirus capsids (*Kristensen and Belsham, 2019*). Our data showed a higher cost to mutation in this motif relative to other capsid positions (*Figure 4—figure supplement 1*), highlighting its importance for capsid function. Finally, we examined the sequence preferences surrounding the two 3CD^pro cleavage sites. We find a strong dependence on the cleavage site residues (positions P1 and P1'; *Figure 5*) and to a lesser degree position P4, with large variation in the sequence preferences across the remaining positions between the two cleavage sites. Overall, our experimentally measured MFE are congruent with existing information regarding the sequence preferences of the

examined capsid motifs, yet provide in-depth insights into sequence specificity that cannot be obtained from examining natural sequence variation.

Finally, we used the amino acid preferences observed in 3CD$^{pro}$ cleavage sites within the capsid to query the human genome for potential cellular targets of this protease (*Figure 5D*). Using this approach, we identify 746 cytoplasmic proteins that harbor a potential 3CD$^{pro}$ target sequence, including 11 proteins previously shown to be cleaved by different picornavirus 3C proteases. We then validated our approach using eight proteins, comprising nine predicted cleavage sites. Six of the predicted cleavage sites were not affected by 3CD$^{pro}$ expression (*Figure 5—figure supplement 1*). On the other hand, three proteins were observed to be specifically cleaved by the viral protease (*Figure 5E*): WD repeat domain 33 (WDR33), an important factor for polyadenylation of cellular pre-mRNAs (*Chan et al., 2014*) that has been shown to act as a restriction factor during influenza infection (*Brass et al., 2009*); the interferon-induced protein phospholipid scramblase 1 (PLSCR1), which is involved in the replication of numerous viruses, likely due to its ability to enhance the expression of certain interferon-stimulated genes (*Kodigepalli et al., 2015*); and the interferon-induced Pleck-strin homology domain containing A4 (PLEKHA4), a plasma membrane-localized signaling modulator (*Shami Shah et al., 2019*) that is currently not known to play a role in viral infection. Overall, our approach correctly predicts 30% of the identified cleavage sites. It is likely that incorporating additional selection criteria, such as accessibility of the cleavage peptide in the folded structure, can be used to further reduce false positives. Nevertheless, extrapolating our validation results to the larger dataset suggests >200 new host targets of the protease are identified, some of which could play key roles in viral biology and pathogenesis.

# Materials and methods

## Key resources table

| Reagent type (species) or resource | Designation | Source or reference | Identifiers | Additional information |
|---|---|---|---|---|
| Strain, strain background (coxsackievirus B3) | pCVB3-XhoI-P1-Kpn21 | 10.1016/j.celrep.2019.09.014 | | Infectious CVB3 clone based on the Nancy strain (Taxon identifier 103903) |
| Strain, strain background (coxsackievirus B3) | pCVB3-XhoI-ΔP1-Kpn21 | This paper | | Infectious CVB3 clone without P1 region |
| Strain, strain background (coxsackievirus B3) | Marked reference CVB3 virus | 10.1038/nmicrobiol.2017.88 | | Infectious CVB3 clone with silent mutations in the polymerase region used as a reference for fitness assays |
| Strain, strain background (*Escherichia coli*) | NZY5α | NZY Tech | MB004 | Competent cells, standard cloning |
| Strain, strain background (*Escherichia coli*) | MegaX DH10B T1R Electrocomp cells | ThermoFisher | C6400-03 | Electrocompetent cells, library cloning |
| Cell line (*Homo sapiens*) | HeLa-H1 | ATCC | CRL-1958; RRID:CVCL_3334 | Cell line for CVB3 infection and DMS library production |
| Cell line (*Homo sapiens*) | HEK293 | ATCC | CRL-1573; RRID:CVCL_0045 | Cell line used for production of CVB3 mutants and for protease cleavage |
| Antibody | Anti-GFP (Mouse monoclonal) | SantaCruz | Sc-9996 | Western blot (1:2000) |

*Continued on next page*

Continued

| Reagent type (species) or resource | Designation | Source or reference | Identifiers | Additional information |
|---|---|---|---|---|
| Antibody | Anti-FLAG (Mouse monoclonal) | SantaCruz | Sc-166335 | Western blot (1:2000) |
| Antibody | Anti-HA (Mouse monoclonal) | SantaCruz | Sc-7392 | Western blot (1:2000) |
| Antibody | Anti-WDR33 (Mouse monoclonal) | SantaCruz | Sc-374466 | Western blot (1:1000) |
| Antibody | Anti-TSG101 (Mouse monoclonal) | SantaCruz | Sc-136111 | Western blot (1:1000) |
| Antibody | Anti-GAK (Mouse monoclonal) | SantaCruz | Sc-137053 | Western blot (1:1000) |
| Antibody | Anti-MAGED1 (Mouse monoclonal) | SantaCruz | Sc-393291 | Western blot (1:1000) |
| Recombinant DNA reagent | DMS libraries (1–3) | This paper | | CVB3 infectious clone libraries with mutagenized capsid region |
| Recombinant DNA reagent | pUC19-HiFi-P1 (plasmid) | This paper | | CVB3 capsid region used as template for DMS cloned into SalI digested pUC19 vector. Used for site-directed mutagenesis |
| Recombinant DNA reagent | T7 encoding plasmid (plasmid) | 10.1128/jvi.02583–14 | RRID:Addgene_65974 | Plasmid encoding T7 polymerase for transfection |
| Recombinant DNA reagent | pIRES-3CDpro (plasmid) | This paper | | CVB3 3 CD protease region cloned into XhoI and NotI pIRES plasmid (Clonetech) |
| Recombinant DNA reagent | peGFP_PLEKHA4 | 10.1016/j.celrep.2019.04.060 | | Kind gift from Dr. Jeremy Baskin GFP-PLEKHA4 expression plasmid |
| Recombinant DNA reagent | peGFP_PLSCR1 | 10.1371/journal.pone.0005006 | | Kind gift from Dr. Serfe Benichou GFP-PLSCR1 expression plasmid |
| Recombinant DNA reagent | pAcGFP-C1 WDR33 | https://doi.org/10.1016/j.molcel.2018.11.036 | | Kind gift from Dr. Matthias Altmeyer pAcGFP-C1 WDR33 expression plasmid |
| Recombinant DNA reagent | FLAG-NLCR5 | Addgene | RRID:Addgene_37521 | NLCR5 expression plasmid |
| Recombinant DNA reagent | HA-ZC3HAV1 | Addgene | RRID:Addgene_45907 | HA-ZC3HAV1 expression plasmid |
| Recombinant DNA reagent | Fluc-eGFP | Addgene | RRID:Addgene_90170 | Fluc-eGFP expression plasmid |

*Continued on next page*

Continued

| Reagent type (species) or resource | Designation | Source or reference | Identifiers | Additional information |
|---|---|---|---|---|
| Sequence-based reagent | HiFi_F | IDT | PCR primer | For generating PCR to clone libraries and sequencing: CTTTGTTGGGTTT ATACCACTTAGC TCGAGAGAGG |
| Sequence-based reagent | HiFi_R | IDT | PCR primer | For generating PCR to clone libraries and sequencing: CCTGTAGTTCCCCA CATACACTGCTCCG |
| Sequence-based reagent | DMS primers | IDT | PCR primer | Primers spanning the full coding region of the CVB3 capsid to perform codon mutagenesis. Listed in *Supplementary file 1*. |
| Sequence-based reagent | 2045_F | IDT | PCR primer | Primer used for Sanger sequencing. TCGAGTGTTTTTA GTCGGACG |
| Sequence-based reagent | 2143_R | IDT | PCR primer | Primer used for Sanger sequencing. TCGAGTGTTTT TAGTCGGACG |
| Sequence-based reagent | 3450_RT | IDT | PCR primer | Primer used for Sanger sequencing and RT-PCR. TCGAGTGTTTTT AGTCGGACG |
| Sequence-based reagent | qPCR_F | 10.1038/nmicrobiol. 2017.88 | PCR primer | qPCR primer for competition assays. GATCGCATATG GTGATGATGTGA |
| Sequence-based reagent | qPCR_R | 10.1038/nmicrobiol. 2017.88 | PCR primer | qPCR primer for competition assays. AGCTTCAGCGAGT AAAGATGCA |
| Sequence-based reagent | MGB_CVB3_wt | 10.1038/nmicrobiol. 2017.88 | TaqManProbe | qPCR probe for competition assays. 6FAM-CGCATCGTA CCCATGG-TAMRA |
| Sequence-based reagent | MGB_CVB3_Ref | 10.1038/nmicrobiol. 2017.88 | TaqManProbe | qPCR probe for competition assays. HEX-CGCTAGCTA CCCATGG-TAMRA |
| Sequence-based reagent | Q8D_F | IDT | PCR primer | Primer for site-directed mutagenesis: gtatcaacgGAT aagactggg |
| Sequence-based reagent | Q8D_R | IDT | PCR primer | Primer for site-directed mutagenesis: ttgag ctcccattttgctgt |

*Continued*

| Reagent type (species) or resource | Designation | Source or reference | Identifiers | Additional information |
|---|---|---|---|---|
| Sequence-based reagent | K829L_F | IDT | PCR primer | Primer for site-directed mutagenesis: gagaaggcaCTA aacgtgaac |
| Sequence-based reagent | K829L_R | IDT | PCR primer | Primer for site-directed mutagenesis: gtattg gcagagtctaggtgg |
| Sequence-based reagent | K235D_F | IDT | PCR primer | Primer for site-directed mutagenesis: gggtcc aacGATttggtacag |
| Sequence-based reagent | K235D_R | IDT | PCR primer | Primer for site-directed mutagenesis: gga tgcgaccggtttgtccgc |
| Sequence-based reagent | R16G_F | IDT | PCR primer | Primer for site-directed mutagenesis: catga gaccGGActgaatgct |
| Sequence-based reagent | R16G_R | IDT | PCR primer | Primer for site-directed mutagenesis: tgccc cagtcttttgcgttg |
| Sequence-based reagent | K827G_F | IDT | PCR primer | Primer for site-directed mutagenesis: caatacgagGG Ggcaaagaac |
| Sequence-based reagent | K827G_R | IDT | PCR primer | Primer for site-directed mutagenesis: gcaga gtctaggtggtctagg |
| Sequence-based reagent | Q566M_F | IDT | PCR primer | Primer for site-directed mutagenesis: atttcgcagATGaacttttc |
| Sequence-based reagent | Q566M_R | IDT | PCR primer | Primer for site-directed mutagenesis: gaaaggagtgt ccttcaatag |
| Sequence-based reagent | T315P_F | IDT | PCR primer | Primer for site-directed mutagenesis: attacgg tcCCCatagcccca |
| Sequence-based reagent | T315P_R | IDT | PCR primer | Primer for site-directed mutagenesis: tgggacgtacgtggtgga |
| Sequence-based reagent | N395H_F | IDT | PCR primer | Primer for site-directed mutagenesis: gagaaggtcCAT tctatggaa |

*Continued on next page*

Continued

| Reagent type (species) or resource | Designation | Source or reference | Identifiers | Additional information |
|---|---|---|---|---|
| Sequence-based reagent | N395H_R | IDT | PCR primer | Primer for site-directed mutagenesis: tccaacatttt ggactgggac |
| Sequence-based reagent | T849A_F | IDT | PCR primer | Primer for site-directed mutagenesis: actacaatgGTC aatacgggc |
| Sequence-based reagent | T849A_R | IDT | PCR primer | Primer for site-directed mutagenesis: gatgctttgcct agtagtgg |
| Sequence-based reagent | K235D_F | IDT | PCR primer | Primer for site-directed mutagenesis: gggtccaacGAT ttggtacag |
| Sequence-based reagent | K235D_R | IDT | PCR primer | Primer for site-directed mutagenesis: ggatgcgacc ggtttgtccgc |
| Sequence-based reagent | 3C_For | IDT | PCR primer | Primer for cloning CVB3 3 CD into pIRES: TATTCTCGAGACC ATGGGCCCTGC CTTTGAGTTCG |
| Sequence-based reagent | 3D_Rev | IDT | PCR primer | Primer for cloning CVB3 3 CD into pIRES: TATTGCGGCCGCC TAGAAGGAGTCC AACCATTTCCT |
| Commercial assay or kit | NEBuilder HiFi DNA Assembly kit | NEB | E2621X | Seamless cloning |
| Commercial assay or kit | TranscriptAid T7 High Yield Transcription Kit | ThermoFisher Scientific | K0441 | T7 in vitro transcription kit |
| Commercial assay or kit | *Quick*-RNA Viral kit | Zymo Research | R1035 | RNA purification |
| Commercial assay or kit | *DNA Clean andConcentrator-5* | Zymo Research | D4013 | DNA purification, gel purification |
| Commercial assay or kit | Luna Universal Probe One-Step RT-qPCR kit | NEB | E3006X | One-step qPCR master mix |
| Chemical compound, drug | Rupintivir | Tocris Biosciences | Cat. #: 6414 | CVB3 3C protease inhibitor |

*Continued on next page*

*Continued*

| Reagent type (species) or resource | Designation | Source or reference | Identifiers | Additional information |
|---|---|---|---|---|
| Software, algorithm | Codon TilingPrimers | https://doi.org/10.1016/j.chom.2017.05.003 | | Software to design primers for mutagenesis (https://github.com/jbloomlab/CodonTilingPrimers) |
| Software, algorithm | Sanger Mutant Library Analysis | Dr. Jesse Bloom | | Software to assess library mutagenesis by Sanger sequencing (https://github.com/jbloomlab/SangerMutantLibraryAnalysis) |
| Software, algorithm | Samtools | http://www.htslib.org/ | version 1.5 | Suite of programs for interacting with high-throughput sequencing data |
| Software, algorithm | Fastp | 10.1093/bioinformatics/bty560 | | Software for NGS read trimming and QC |
| Software, algorithm | PicardTools, FastqToSam | https://broadinstitute.github.io/picard/ | Version 2.2.4 | Used to generate Bam files from Fastq files |
| Software, algorithm | Duplex pipeline | https://github.com/KennedyLabUW/Duplex-Sequencing; *Kennedy et al., 2014* | Version 3.0 | Analysis pipeline for duplex sequencing (UnifiedConsensusMaker.py) |
| Software, algorithm | VariantBam | 10.1093/bioinformatics/btw111 | | Software to filter Bam files |
| Software, algorithm | BWA | https://sourceforge.net/projects/bio-bwa/files/ | Version 0.7.16 | Software to align NGS reads |
| Software, algorithm | Fgbio | http://fulcrumgenomics.github.io/fgbio/ | version 1.1.0 | Software used to hard-clip NGS reads |
| Software, algorithm | VirVarSeq | 10.1093/bioinformatics/btu587 | version 1.1.0 | Software used to identify codons in each NGS read |
| Software, algorithm | Custom R scripts | This paper | | Custom R scripts to process output of VirVarSeq script. Available at https://github.com/RGellerLab/CVB3_Capsid_DMS |
| Software, algorithm | DMS_tools2 | 10.1186/s12859-015-0590-4 | | Software to determine amino acid preferences and mutational fitness effects |
| Software, algorithm | TANGO | 10.1038/nbt1012 | | Software to determine the effect of mutations on aggregation |
| Software, algorithm | FoldX | 10.1093/nar/gki387 | | Software to determine the effect of mutations on stability |

*Continued on next page*

*Continued*

| Reagent type (species) or resource | Designation | Source or reference | Identifiers | Additional information |
|---|---|---|---|---|
| Software, algorithm | DSSP | http://swift.cmbi.ru.nl/gv/dssp/ | | Software used to obtain secondary structure and RSA within DMS_tools2 |
| Software, algorithm | ViprDB | http://viperdb.scripps.edu/ *Carrillo-Tripp et al., 2009* | | Software used to obtain structural information on capsid sites |
| Software, algorithm | DECIPHER Package | 10.32614/RJ-2016–025 | | R package for performing codon alignments |
| Software, algorithm | PhyDMS | doi: 10.7717/peerj.3657 | | For phylogenetic and differential selection analyses. https://jbloomlab.github.io/phydms/index.html |
| Software, algorithm | Custom R scripts | This paper | | Custom R script to generate in silico peptides spanning 10AA 3 CD protease cleavage site. Available at https://github.com/RGellerLab/CVB3_Capsid_DMS |
| Software, algorithm | PSSMSearch | 10.1093/nar/gky426 | | Used to generate position-specific scoring matrix and search human proteome for hits. http://slim.icr.ac.uk/pssmsearch/ |
| Software, algorithm | Peptides R package | ISSN 2073–4859 | Version 2.4.2 | R package to predict molecular weight of proteins |
| Software, algorithm | RandomForest R package | 10.1023/A:1010933404324 | Version 4.6–16 | R package for random forest prediction |
| Software, algorithm | Logolas | 10.1186/s12859-018-2489-3 | | Package to generate logo plots in R |

## Viruses, cells, and plaque assays

HeLa-H1 (CRL-1958; RRID:CVCL_3334) and HEK293 (CRL-1573; RRID:CVCL_0045) cells were obtained from ATCC and were periodically validated to be free of mycoplasma. All work with CVB3 was based on the Nancy infectious clone (kind gift of Dr. Marco Vignuzzi, Institute Pasteur). Cells were cultured in culture media (Dulbecco's modified Eagle's medium [DMEM] with 10% heat-inactivated fetal bovine serum (FBS), Pen-Strep, and L-glutamine) with FBS concentrations of 2% during infection. For plaque assays, serial dilutions of the virus were used to infect confluent HeLa-H1 cells in six-well plates for 45 min, followed by overlaying the cells with a 1:1 mixture of 56°C 1.6% agar (Arcos Organics 443570010) and 37°C 2× DMEM with 4% FBS. Two days later, plates were fixed with formaldehyde (2% final concentration) after which the agar was removed and the cells stained with crystal violet to visualize plaques.

## Deep mutational scanning

The infectious clone was modified by site-directed mutagenesis to remove an XhoI site present in the capsid region (P1) and introduce an XhoI site at position 692 as well as a Kpn2I site at position 3314, generating a pCVB3-XhoI-P1-Kpn2I clone (*Bou et al., 2019*). In addition, a pCVB3-XhoI-ΔP1-Kpn2I plasmid was generated by replacing the region between the XhoI and Kpn2I sites in pCVB3-XhoI-P1-Kpn2I with a short linker. To generate the template for DMS, the capsid region was amplified by PCR from pCVB3-XhoI-P1-Kpn2I with Phusion polymerase (Thermo Scientific) and primers HiFi-F (CTTTGTTGGGTTTATACCACTTAGCTCGAGAGAGG) and HiFi-R (CCTGTAGTTCCCCACA TACACTGCTCCG) and gel purified (Zymoclean Gel DNA Recovery Kit). Primers spanning the full coding region of the capsid region were designed using the CodonTilingPrimers software from the Bloom lab (https://github.com/jbloomlab/CodonTilingPrimers; *Dingens et al., 2017*) with the default parameters and synthesized by IDT (*Supplementary file 1*). These primers were used to perform the mutagenesis PCR on the capsid template together with the HiFi-F or HiFi-R primers in triplicate following published protocols (*Dingens et al., 2017*) with the exception that 10 rounds of mutagenesis were performed for libraries 1 and 2, while a second round of seven mutagenesis cycles was performed for library three to increase the number of mutation per clone. The products were gel purified and ligated to an XhoI and Kpn2I digested and gel purified pCVB3-XhoI-ΔP1-Kpn2I using NEBuilder HiFi DNA Assembly reaction (NEB) for 25 min. Mutagenesis efficiency was evaluated by the transformation of the assembled plasmids into NZY5α competent cells (NZY Tech), Sanger sequencing of 18–23 clones per library, and mutation analysis using the Sanger Mutant Library Analysis script (https://github.com/jbloomlab/SangerMutantLibraryAnalysis; *Bloom, 2014*). Subsequently, the assembled plasmid reactions were purified using a Zymo DNA Clean and Concentrator-5 kit (Zymo Research) and used to electroporate MegaX DH10B T1R Electrocomp cells (Thermo-Fisher) using a Gene Pulser XCell electroporator (Bio-Rad) according to the manufacturer's protocol. Cells were then grown overnight in a 50 mL liquid culture at 33°C and DNA purified using the Pure-Link HiPure plasmid midiprep kit (Invitrogen). Transformation efficiency was estimated by plating serial dilutions of the transformation on agar plates. In total, $4.44 \times 10^5$, $1.46 \times 10^5$, and $2.19 \times 10^5$ transformants were obtained for lines 1, 2, and 3, respectively. Viral genomic RNA was then transcribed from SalI linearized, gel-purified full-length plasmids using the TranscriptAid T7 kit (Thermo Scientific), and four electroporations were performed using $4 \times 10^6$ HeLa-H1 cells in a 4 mm cuvette in 400 µL of calcium- and magnesium-free phosphate-buffered saline (PBS) using with 8 µg of RNA in a Gene Pulser XCell (Bio-Rad) set to 240 V and 950 µF. Electroporated cells were then pooled, and one-fourth was cultured for 9 hr to produce the passage 0 virus (P0). Following three freeze–thaw cycles, $2 \times 10^6$ plaque-forming units (PFU) were used to infect a 90% confluent 15 cm plate in 2.5 mL of infection media for 1 hr. Cells were then washed with PBS and incubated in 12 mL of infection media for 9 hr. Finally, cells were subjected to three freeze–thaw cycles, debris removed by centrifugation at $500 \times$ g, and the supernatants collected to generate P1 virus stocks. All infections produced $>2.38 \times 10^6$ PFU in P0 and $>1.2 \times 10^7$ PFU in P1 as judged by plaque assay.

## Next-generation sequencing analysis

Libraries were prepared following published protocols (*Kennedy et al., 2014*), and each library was run on a Novaseq6000 2 × 150 at a maximum of 30G per lane to reduce potential index hopping. Reads trimming was performed using fastp (*Chen et al., 2018*) (command: –max_len1 150 –max_-len2 150 –length_required 150 -x -Q -A), unsorted bam files were generated from fastq files using Picard tools FastqToSam (version 2.2.4) and merged into a single bam using the cat command of Samtools (version 1.5). The duplex pipeline was then implemented (https://github.com/KennedyLa-bUW/Duplex-Sequencing/UnifiedConsensusMaker.py; *Kennedy et al., 2014*). using the UnifiedConsensusMaker.py script and a minimum family size of 3, a cutoff of 0.9 for consensus calling, and an N cutoff of 0.3. The single-stranded consensus files (SSCS) were then aligned using BWA mem (version 0.7.16), sorted using Samtools, size selected to be 133 bp long using VariantBam (*Wala et al., 2016*), unaligned reads were discarded (Samtools view command with -F 4), and the resulting bam file indexed with Samtools. Subsequently, fgbio (http://fulcrumgenomics.github.io/fgbio/; version 1.1.0) was used to hard-clip 10 bp from each end and upgrade all clipping to hard-clip (-c Hard –upgrade-clipping true –read-one-five-prime 10 –read-one-three-prime 10 –read-two-five-prime 10 –read-two-three-prime 10). Variant bam was then used to keep all reads

that were between 50 and 150 bp, well-mapped, and had either no indels and less than five mutations (command –r {'':{'rules':[{'ins':[0,0],'del':[0,0],'nm':[0,4], 'mate_mapped':true,'fr':true,'length': [50,150]}]}}"). Finally, the codons in each read were identified using the VirVarSeq (*Verbist et al., 2015*) Codon_table.pl script using a minimum read quality of 20. A custom R script was then used to generate a codon counts table for each codon position by eliminating all codons containing ambiguous nucleotides and codons with a strong strand bias (StrandOddsRatio > 4), as well as all codons that are reached via a single mutation (available at https://github.com/RGellerLab/CVB3_Capsid_DMS); *Mattenberger, 2021*; copy archived at swh:1:rev:29dd205182f0886dc5bad3e6b4dd-d6e786c58a75). Amino acid preferences and MFE were determined using DMStools2 (*Bloom, 2015*), with the Bayesian option and the default settings.

## Structural analyses

The crystal structure PDB:4GB3 (*Yoder et al., 2012*) was used for all structural analyses. The effects of mutations on aggregation were determined using TANGO version 2.3.1 (*Fernandez-Escamilla et al., 2004*) using the default settings, and the effect on stability on the monomer and pentamer was determined using FoldX 4 (*Schymkowitz et al., 2005*) using the default settings. For the latter, the pentamer subunits were renamed to unique letters, all mutations between the reference sequence and the structure sequence were introduced using the BuildModel command, the structure was optimized using the RepairPDB command 5 or 10 times for the pentamer or monomer, respectively, and then the effects of the mutations were predicted using the BuildModel command (modified PDB files can be found at https://github.com/RGellerLab/CVB3_Capsid_DMS). Secondary structure and RSA were obtained from DSSP (http://swift.cmbi.ru.nl/gv/dssp/) using the dms_tools2. dssp function of dms_tools2, while interface, surface, and core residues as well as residue contact number, and presence in the twofold, threefold, and fivefold axes were obtained from ViprDB (http://viperdb.scripps.edu/) (*Carrillo-Tripp et al., 2009*). Distance from the center was calculated with Pymol using the Distancetoatom.py script on the monomer or pentamer. Finally, the location of antibody neutralization sites in CVB3 was obtained from an analysis of the CVB3 capsid structure in a previous publication (*Muckelbauer et al., 1995*).

## Generation and evaluation of CVB3 capsid mutants

With the exception of mutant N395H (kind gift of Rafael Sanjuan) (*Bou et al., 2019*), all other mutants were generated by site-directed mutagenesis. For this, the PCR of the capsid region used as a template for DMS was phosphorylated and cloned into a SmaI digested pUC19 vector for use in the mutagenesis reactions (pUC19-HiFi-P1). For each mutant, non-overlapping primers containing the mutation in the middle of the forward primer were used to introduce the mutation with Phusion polymerase, followed by DpnI (Thermo Scientific) treatment, phosphorylation, ligation, and transformation of chemically competent bacteria. Successful mutagenesis was verified by Sanger sequencing. Subsequently, the capsid region was subcloned into pCVB3-XhoI-ΔP1-Kpn2I using XhoI and Kpn2I sites. Plasmids were then linearized with MluI, and 2 μg of plasmid was transfected into 5 × $10^5$ HEK293 cells, together with a plasmid encoding the T7 polymerase (*Yun et al., 2015*) (Addgene 65974) using calcium phosphate. Briefly, an equal volume of 2× HBS (274 mM NaCl, 10 mM KCl, 1.4 mM $Na_2HPO_4$) was added dropwise to DNA containing 0.25M $CaCl_2$ while mixing, incubated 15 min at RT, and then added dropwise to cells. Following 48 hr, passage 0 (P0) virus was collected and titered by plaque assay. From this, $10^5$ PFU were used to infect 90% confluent six-well HeLa-H1 cells (multiplicity of infection (MOI) 0.1) for 1 hr at 37°C, after which the cells were washed twice with PBS and 2 mL of infection media added. Cells were then incubated until cytopathic effect (CPE) was observed. Emerging viral populations were titered by plaque assay and the capsid region sequenced to ensure no compensatory mutations or reversions arose during replication. The fitness of these mutants was then tested by direct competition with a marked reference virus using a Taqman RT-PCR method (*Moratorio et al., 2017*). Briefly, using four biological replicates, confluent HeLa-H1 cells in a 24-well plate were infected with 200 μL of a 1:1 mixture of 4 × $10^3$ PFU (MOI 0.01) of the test and marked reference viruses for 45 min. Subsequently, the inoculum was removed, the cells were washed twice with PBS, 200 μL of infection media was added, and the cells were incubated for 24 hr at 37°C. Finally, cells were subjected to three freeze–thaw cycles, debris removed by centrifugation at 500 × g, the supernatants collected and treated with 2 μL of RNase-Free DNaseI

(ThermoFisher) for 15 min at 37°C, and viral RNA extracted using the *Quick*-RNA Viral Kit (Zymo Research), eluting in 20 µL. Quantification of the replication of each mutant versus the reference was performed using Luna Universal Probe One-Step RT-qPCR kit (New England BioLabs) containing 3 µL of total RNA, 0.4 µM of each qPCR primers, and 0.2 µM of each probe. The standard curve was performed using 10-fold dilutions of RNA extracted from $10^7$ PFU of wild-type and reference viruses. All samples were performed with three technical replicates. The relative fitness (W) of each mutant versus the common marked reference virus was calculated using the following formula: $W = [R(t)/R(0)]^{1/t}$, where $R(0)$ and $R(t)$ represent the ratio of the mutant to the reference virus genomes in the initial mixture used for the infection and after 1 day (t = 1), respectively (*Carrasco et al., 2007*; *Moratorio et al., 2017*).

## Sequence variability and phylogenetic analyses

Amino acid variability was assessed using Shannon entropy. Briefly, all available, non-identical, full-genome CVB3, CVB, or enterovirus B sequences were downloaded from Virus Pathogen Resource (*Pickett et al., 2012*) (http://www.viprbrc.org) and codon-aligned using the DECIPHER package in R (available at https://github.com/RGellerLab/CVB3_Capsid_DMS). All alignment positions not present in our reference strain were removed, and a custom R script was used to calculate Shannon entropy. For phylogenetic and differential selection analyses, PhyDMS was run using the default settings on an alignment of CVB3 genomes that was processed with the phydms_prealignment module and using the average preferences from the three DMS replicates.

## Identification of 3CD^pro cleavage sites in the human proteome

The amino acid preferences (the relative enrichment of each amino acid at each position standardized to 1) was used to generate in silico 1000 peptides spanning the 10 amino acid regions surrounding each cleavage site using a custom R script (available at https://github.com/RGellerLab/CVB3_Capsid_DMS). Specifically, for each peptide position, 100 peptides were generated that encoded each amino acid at a frequency corresponding to its preference observed in the DMS results, with the remaining positions unchanged. The resulting 1000 peptides from each cleavage site were uploaded to PSSMSearch (*Krystkowiak et al., 2018*) (http://slim.icr.ac.uk/pssmsearch/) using the default setting (psi_blast IC). Results were filtered to remove proteins indicated to be secreted, lumenal, or extracellular in the Warnings column. To test whether proteins were cleaved by the viral 3 CD protease, the corresponding region was PCR amplified from the Nancy infectious clone (primers 3C-For: TATTCTCGAGACCATGGGCCCTGCCTTTGAGTTCG and 3D-Rev: TATTGCGGCCGCC TAGAAGGAGTCCAACCATTTCCT) and cloned into the pIRES plasmid (Clonetech) using the restriction sites XhoI and NotI (pIRES-3CD^pro). For analysis of fusion proteins, HEK293 cells were transfected with GFP-PLEKHA4 (kind gift of Dr. Jeremy Baskin, Cornell University), GFP-PLSCR1 (kind gift of Dr. Serge Benichou, Institut Cochin), pAcGFP-WDR33 (Kind gift of Dr. Matthias Altmeyer, University of Zurich), FLAG-NLCR5 (Addgene #37521), HA-ZC3HAV1 (Addgene #45907), or the control plasmid FLuc-eGFP (Addgene #90170), together with the pIRES-3CD^pro plasmid using Lipofectamine 2000. Following 24 hr, proteins were collected by lysing in lysis buffer (50 mM Tris–HCl, 150 mM NaCl, 1% NP40, and protease inhibitor cocktail [Complete Mini EDTA-free, Roche]) and subjected to western blotting with the corresponding antibody (anti-GFP, Santa Cruz sc-9996; anti–FLAG, Santa Cruz sc-166335; anti-HA, Santa Cruz, sc-7392). For analysis of endogenous proteins, 3CD^pro was expressed for 48 hr before cell lysis, and western blotting using antibodies against WDR33 (Santa Cruz sc-374466), TSG101 (Santa Cruz sc-136111), GAK (Santa Cruz sc-137053), and MAGED1 (Santa Cruz sc-393291). When indicated, the 3C^pro inhibitor rupintrivir (Tocris Biosciences) was added at a concentration of 2 µM for the last 24 hr before collection. The predicted molecular weight of cleaved fragments was calculated using the mw function of the Peptides R package (version 2.4.2).

## Statistical analyses

All experiments were performed with at least three biological replicates with the exception of the analysis of protein cleavage by western blotting, which was performed in duplicate. All statistical analyses were performed in R and were two tailed. For random forest prediction, the R RandomForest package (version 4.6–14) was employed using the default setting with an mtry of 10, and for the linear model, the formula lm(MFE ~ enterovirus B entropy + WT amino acid * mutant amino acid +

predicted effect of mutations on stability in the pentamer + relative surface exposure) was used (available at https://github.com/RGellerLab/CVB3_Capsid_DMS). Sequence logoplots were producing using Logolas (*Dey et al., 2018*).

## Data availability

Unaligned bam files have been uploaded to SRA (Bioproject PRJNA643896, SRA SRP269871, Accession SRX8663374-SRX8663384). The scripts and data required to obtain the codon count tables for all samples, to perform the random forest and linear model predictions, to generate the peptides for use with PSSMsearch, as well as the sequence alignments and modified structure files for FoldX analysis, can be found on GitHub (https://github.com/RGellerLab/CVB3_Capsid_DMS). Finally, the interactive heatmap of MFE across the capsid was generated by modifying a script from a prior publication (*Starr et al., 2020*) (available at https://github.com/jbloomlab/SARS-CoV-2-RBD_DMS/blob/master/interactive_heatmap.ipynb) and can be found on this projects' GitHub page (https://github.com/RGellerLab/CVB3_Capsid_DMS).

## Acknowledgements

The authors would like to thank Dr. Javier O Cifuentes for help with the interpretation of antibody neutralization sites and Drs. Santiago Elena and Tzachi Hagai for critical reading of the manuscript. In addition, the authors would like to acknowledge the use of the Principe Felipe Research Center (CIPF) server which was co-financed by the European Union through the Operativa Program of the European Regional Development Fund (ERDF/FEDER) of the Comunitat Valenciana 2014–2020.

## Additional information

### Funding

| Funder | Grant reference number | Author |
| --- | --- | --- |
| Ministerio de Economía, Industria y Competitividad, Gobierno de España | BFU2017-86094-R | Ron Geller |
| Ministerio de Economía, Industria y Competitividad, Gobierno de España | RYC-2015-17517 | Ron Geller |
| Ministerio de Economía, Industria y Competitividad, Gobierno de España | BES-2016-076677 | Florian Mattenberger |

The funders had no role in study design, data collection and interpretation, or the decision to submit the work for publication.

### Author contributions

Florian Mattenberger, Conceptualization, Formal analysis, Investigation, Methodology, Writing - original draft, Writing - review and editing; Victor Latorre, Conceptualization, Investigation, Methodology, Writing - review and editing; Omer Tirosh, Software, Formal analysis, Methodology; Adi Stern, Software, Formal analysis, Supervision, Methodology, Writing - original draft, Writing - review and editing; Ron Geller, Conceptualization, Data curation, Formal analysis, Supervision, Funding acquisition, Investigation, Methodology, Writing - original draft, Writing - review and editing

### Author ORCIDs

Florian Mattenberger (iD) https://orcid.org/0000-0002-2727-0284
Omer Tirosh (iD) https://orcid.org/0000-0001-8139-9866
Adi Stern (iD) http://orcid.org/0000-0002-2919-3542
Ron Geller (iD) https://orcid.org/0000-0002-7612-4611

Decision letter and Author response
Decision letter https://doi.org/10.7554/eLife.64256.sa1
Author response https://doi.org/10.7554/eLife.64256.sa2

## Additional files

### Supplementary files
- Supplementary file 1. Primers for next-generation sequencing.
- Supplementary file 2. Next-generation sequencing statistics.
- Supplementary file 3. Mutations observed by Sanger sequencing.
- Supplementary file 4. Mutational fitness effects of the mutagenized viral populations.
- Supplementary file 5. Results of qPCR validation of MFE.
- Supplementary file 6. Data used for random forest model and parameter explanation.
- Supplementary file 7. Differential selection results.
- Supplementary file 8. PSSMsearch results.
- Transparent reporting form

### Data availability
Sequencing data have been uploaded to SRA (Bioproject PRJNA643896, SRA SRP269871, Accession SRX8663374-SRX8663384). All data used in the paper are either included as supplemental data and/or can be found at https://github.com/RGellerLab/CVB3_Capsid_DMS (copy archived at https://archive.softwareheritage.org/swh:1:rev:29dd205182f0886dc5bad3e6b4ddd6e786c58a75/).

The following dataset was generated:

| Author(s) | Year | Dataset title | Dataset URL | Database and Identifier |
|---|---|---|---|---|
| Mattenberger F, Geller R | 2020 | Deep mutational scanning of the CVB3 capsid protein | https://www.ncbi.nlm.nih.gov/bioproject/643896 | NCBI BioProject, PRJNA643896 |

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
