## [Decision Letter]

**Acceptance summary:**

In this manuscript state of the art techniques are used to systematically mutate the viral genome and measure its effect on viral fitness. This paper provides novel insights in viral pathogenesis and evolution of an important class of viruses.

**Decision letter after peer review:**

Thank you for submitting your article "Globally defining the effects of mutations in a picornavirus capsid" for consideration by *eLife*. Your article has been reviewed by two peer reviewers, one of whom is a member of our Board of Reviewing Editors, and the evaluation has been overseen by Sara Sawyer as the Senior Editor. The following individual involved in review of your submission has agreed to reveal their identity: Jesse D Bloom (Reviewer #2).

The reviewers have discussed the reviews with one another and the Reviewing Editor has drafted this decision to help you prepare a revised submission.

This paper will be of interest to scientists studying viral pathogenesis and evolution. The authors use state of the art techniques to systematically mutate the viral genome and measure its effect on viral fitness. The results are robust and support the key claims in the paper.

Summary:

In this manuscript Geller and colleagues describe a deep mutational scan of the coxsackie B3 virus (CV-B3) genome focusing on the structural genes. Comparing the mutations before and after growth of the virus in cell culture allows the authors to derive the mutational fitness effect (MFE) of each mutated residue. After generating this valuable and robust experimental dataset, they compare the MFE with variability observed in strains directly sequenced in nature and observe a correlation suggesting that the experimental fitness in the lab is representative of natural evolutionary processes. They also identify sites under differential selection between the laboratory selection and natural selection, which correspond to known antibody neutralization sites. Using a random forest algorithm the authors show that a combination of evolutionary, sequence and structural information best explains MFE. Finally they focus on the myristoylation and protease cleavage site. The deep mutagenesis allowed for refinement of the 3CDpro cleavage consensus site and allowed for the identification of ~750 cellular proteins that could be substrate. Testing a subset of these, the authors show that 30% indeed could be experimentally verified.

Overall, this well-written manuscript provides an important resource through deep mutagenesis of the picornavirus capsid. The analysis of this dataset and the experimental follow up performed by the authors has generated novel insights in picornavirus structure and function. The study is very nicely done. The data will be of interest to researchers interested in these techniques, as well as researchers interested in picornaviruses themselves. There are also some nice evolutionary and structural analyses, as well as cool use of the deep mutational scanning to identify some new host proteins cut by the viral protease.

1) The authors validate the 3CDpro cleavage sites through expression of 3CDpro outside of the context of viral infection. Looking at the cleavage of the candidate substrates during infection with CV-B3 will be more relevant and perhaps increase the percentage of candidates that can be cleaved by 3CDpro. (Although this would improve the manuscript bolstering the physiological relevance of these candidate substrates, this suggestion is completely optional and not required for acceptance of the manuscript).

2) It would be interesting to extend the analysis to also include the 2A cleavage site. Because only the structural genes are mutated, the analysis can only be done for the site before the protease cleavage residue but might be interesting regardless.

3) Some quite relevant data is now in the supplementary files, which might be missed by the readers. For example the validation with qPCR in Supplementary file 5. Perhaps the result can be also graphed as XY graph showing the correlation in the main figure.

4) The authors refer to "antibody neutralization sites", but they never explain clearly how these were defined. Maybe this is common knowledge in the picornavirus field, but it certainly wasn't obvious to me and would benefit from some explanation. I gather these are sites where mutations escape some antibody?

5) In Figure 2 and some of the surrounding text in the Results (such as the prediction section), it is difficult to tell if the authors are using MFE to refer to the effects of specific mutations, or the mean effects of mutations at each site. This could be better explained for each relevant analysis.

6) Be sure to include adequate text to emphasize these are effects of mutations in cell culture, which might not always mirror effects in nature.

7) The use of the DMS data to make a PSSM to find other proteins cleaved by the protease is cool!

8) Introduction, last paragraph: should be "these data", not "this data".

9) When the authors start using the word "fitness," they should be explicit that this is "fitness" as measured by growth in cell culture, which may not always be precisely the same as fitness in nature.

10) Figure 2A, would be nice if legend is a bit clearer about what "average MFE" (red line) means: I gather average across mutations. Is the average MFE also windowed in 21 amino-acid windows?

11) This is a *completely optional* suggestion that the authors can feel free to ignore. But we found in our recent DMS that making an interactive heat map (e.g., https://jbloomlab.github.io/SARS-CoV-2-RBD_DMS/) was really useful for enabling people to interrogate the data. These can be made pretty easily using Altair (https://altair-viz.github.io/), and some example code is here (https://github.com/jbloomlab/SARS-CoV-2-RBD_DMS/blob/master/interactive_heatmap.ipynb). Yours would actually be a lot simpler to make as there is just one phenotype shown.

---

## [Author Response]

1) The authors validate the 3CDpro cleavage sites through expression of 3CDpro outside of the context of viral infection. Looking at the cleavage of the candidate substrates during infection with CV-B3 will be more relevant and perhaps increase the percentage of candidates that can be cleaved by 3CDpro. (Although this would improve the manuscript bolstering the physiological relevance of these candidate substrates, this suggestion is completely optional and not required for acceptance of the manuscript).

We agree that evaluating protein cleavage in the context of infection is important and can potentially impact cleavage efficiency. However, performing the experiments in the context of infection complicates the interpretation of the results as cleavage could also be mediated by the second viral protease (2A). We will include this validation once we set up the tools to test 2A cleavage targets (see comment 2 below).

2) It would be interesting to extend the analysis to also include the 2A cleavage site. Because only the structural genes are mutated, the analysis can only be done for the site before the protease cleavage residue but might be interesting regardless.

We completely agree with the reviewers’ comment and are planning on performing this analysis. However, we prefer to wait until we have generated the data for the second half of the protease cleavage site (residues present in the 2A coding region) to be able to analyze the full cleavage site since this can eliminate false positives.

3) Some quite relevant data is now in the supplementary files, which might be missed by the readers. For example the validation with qPCR in Supplementary file 5. Perhaps the result can be also graphed as XY graph showing the correlation in the main figure.

We apologize for not including this data in the primary figures. We have now included the qPCR validation in the main figure (Figure 2G). Please note that for formatting reasons, we have combined the original plots for F and G into a single F panel to accommodate the new figure panel.

4) The authors refer to "antibody neutralization sites", but they never explain clearly how these were defined. Maybe this is common knowledge in the picornavirus field, but it certainly wasn't obvious to me and would benefit from some explanation. I gather these are sites where mutations escape some antibody?

We apologize for failing to explain this in the original text. Indeed, antibody neutralization sites have been defined using different picornaviruses by identifying escape mutants from monoclonal antibodies (mainly using poliovirus) as well as structural analyses of capsid bound to antibodies. The positions of the neutralization sites in the CVB3 capsid were derived from an analysis performed in a publication describing the high-resolution structure of CVB3 (Muckelbauer et al., 1995). We now include this reference/information in the Materials and methods section (last sentence in the Structural analyses section).

5) In Figure 2 and some of the surrounding text in the Results (such as the prediction section), it is difficult to tell if the authors are using MFE to refer to the effects of specific mutations, or the mean effects of mutations at each site. This could be better explained for each relevant analysis.

Again, we apologize for the lack of clarity. We have now included this information in the legend for Figure 2 and the corresponding location in the main text.

6) Be sure to include adequate text to emphasize these are effects of mutations in cell culture, which might not always mirror effects in nature.

We have now included such statements in both the main text and in the Discussion.

7) The use of the DMS data to make a PSSM to find other proteins cleaved by the protease is cool!

Thanks!

8) Introduction, last paragraph: should be "these data", not "this data".

We thank the reviewers for catching this error. We have corrected the error.

9) When the authors start using the word "fitness," they should be explicit that this is "fitness" as measured by growth in cell culture, which may not always be precisely the same as fitness in nature.

We have indicated this in both the main text and in the Discussion.

10) Figure 2A, would be nice if legend is a bit clearer about what "average MFE" (red line) means: I gather average across mutations. Is the average MFE also windowed in 21 amino-acid windows?

Again, we apologize for not making this clear initially. Indeed, this is as you assumed, and we have added additional text in the figure legend to clarify this.

11) This is a completely optional suggestion that the authors can feel free to ignore. But we found in our recent DMS that making an interactive heat map (e.g., https://jbloomlab.github.io/SARS-CoV-2-RBD_DMS/) was really useful for enabling people to interrogate the data. These can be made pretty easily using Altair (https://altair-viz.github.io/), and some example code is here (https://github.com/jbloomlab/SARS-CoV-2-RBD_DMS/blob/master/interactive_heatmap.ipynb). Yours would actually be a lot simpler to make as there is just one phenotype shown.

We have included now a link to this manner of representing the data. Once again, thank you for developing such accessible code and excellent tools. These can be found on our GitHub website, and we reference the original script and the publication from which it was derived in the Materials and methods section.